# Evidence-based guidance on reflective pavement for urban heat mitigation in Arizona

Florian A. Schneider [1] ✉, Johny Cordova Ortiz[2], Jennifer K. Vanos[1], David J. Sailor[3] & Ariane Middel [4,5]

Urban overheating is an increasing threat to people, infrastructure, and the environment. Common heat mitigation strategies, such as green infrastructure, confront space limitations in current car-centric cities. In 2020, the City of Phoenix, Arizona, piloted a "cool pavement" program using a solar reflective pavement seal on 58 km of residential streets. Comprehensive micrometeorological observations are used to evaluate the cooling potential of the reflective pavement based on three heat exposure metrics—surface, air, and mean radiant temperatures—across three residential reflective pavement-treated and untreated neighborhoods. In addition, the solar reflectivity of reflective pavement is observed over 7 months across eight residential neighborhoods. Results are synthesized with the literature to provide context-based reflective pavement implementation guidelines to mitigate urban overheating where common strategies cannot be applied. The three most important contextual factors to consider for effective implementation include urban location, background climate type, and heat exposure metric of interest.

Cities globally have experienced elevated temperatures due to anthropogenic heat sources and the built environment[1], which results in an Urban Heat Island (UHI) by adding or retaining more energy within the urban system[2,3]. This urban-induced warming together with climate change result in increasing temperatures in cities[4]. Urban overheating, a recently introduced concept, is of particular concern within already-hot cities experiencing extensive urban heat stress for their residents, such as the City of Phoenix, Arizona, USA[5,6]. These concerns are intensified due to unequal distribution of heat exposure resulting from past and current marginalization and contemporary urban design decisions[7,8]. For example, neighborhood-level decisions involving heat mitigation strategies affect residential heat exposure and energy use[9–11], which are impacted by environmental racism and historical neighborhood redlining[8,12]. While there has been extensive past focus within the scientific literature on the UHI concept[13], quantifying the magnitude and impact of *intra*-urban heat variability is a more meaningful way to

understand drivers affecting human health, energy use, and water use, among other impacts.

Urban overheating constitutes a multi-faceted threat to the well-being, performance, and health of individuals, as well as the energy efficiency and economy of cities[13]. Numerous heat mitigation strategies have, or will be, deployed and tested to counteract overheating in cities worldwide. Heat mitigation strategies, such as increased tree canopy or urban green spaces, are seen as compelling solutions to cool the urban environment based on various heat exposure metrics (including 2m-air, surface, and mean radiant temperatures) while creating esthetically appealing spaces that include other co-benefits, such as increased shade, property value, recreation space, and ecosystem services[14–17]. However, urban green spaces can only provide these amenities if well-watered, maintained, and accessible[18,19]. Cities in hot and dry environments, such as the City of Phoenix, face water shortages that could affect the ability to maintain trees and grass and thus jeopardize their cooling potential[20]. Additionally, many cities are

[1]School of Sustainability, Arizona State University, Tempe, AZ, USA. [2]School of Sustainable Engineering and the Built Environment, Arizona State University, Tempe, AZ, USA. [3]School of Geographical Sciences and Urban Planning, Arizona State University, Tempe, AZ, USA. [4]School of Arts, Media and Engineering, Arizona State University, Tempe, AZ, USA. [5]School of Computing and Augmented Intelligence, Arizona State University, Tempe, AZ, USA. ✉e-mail: florian.schneider@asu.edu

car-centric, with extensive use of paving. In Phoenix, a high surface area (36%) covered by streets and parking lots[21] makes large-area implementation of urban green spaces more difficult. Paved surfaces have high thermal storage capacities and sensible heat fluxes[22], two major components responsible for the additional urban heat and thus the higher overall heat load.

Reflective coatings are one strategy to reduce surface temperature and heat storage by pavements and roofs[23]. Additionally, a reflective coating may increase road service life under normal operating conditions because asphalt-based pavements wear and develop cracks due to higher surface and internal temperature ranges. Yet, reflective coatings or pavement have not been evaluated as a feasible heat mitigation strategy for car-centric spaces where urban green spaces cannot be implemented.

Reflective coatings have a higher albedo (reflectance across the solar radiation spectrum) and thus reduce solar radiation absorption yielding lower surface temperatures[23–26]. Reflective coatings are generally easy to apply to existing paved surfaces (e.g., spray, squeegee) and, in most cases, use light-colored pigments and materials (such as nanoparticles) to increase albedo[23,27–30]. The technology is a low-cost measure, which is particularly important as the cost-effectiveness of heat mitigation strategies is key to widespread implementation, but tends to be neglected[31]. Reflective coatings are stated to require minimal maintenance[30,32], do not need water to be effective[33] (which is of particular interest to water-strained areas), and can be applied citywide[24], including areas that cannot be used for urban green spaces to provide cooling. City-wide application of reflective pavement (RP) may be restricted to use on roads with certain (lower) speed limits and roads not requiring line striping/traffic paint, following city practice based on industry safety guidance.

A conventional seal coat returns the road surface to a low albedo (~5%) and ages over time to ~12–13% reflectivity, while the reflective coating can be 6–7 times (and more) as reflective as the initial albedo (at 30–35%). Even higher albedos are possible. However, since highly reflective coatings at ground-level (e.g., pavements) may have adverse effects such as glare[34], it is advisable to consider moderately reflective coatings (e.g., with solar reflectance <50%). When applied on roadways for heat mitigation, these coatings are called "cool" or "reflective" pavements (CP and RP, respectively). CP often refers to multiple technologies that create a cooler surface than traditional concrete or asphalt concrete (AC). RP is one of these CP technologies accomplished via coatings[23,25], with further CP technologies including phase-change material pavement[28], highly conductive pavements[28], pavement as solar collectors[35,36], and permeable pavement, i.e., porous pavement[23] and water-retentive pavement[37].

Research on reflective urban materials, particularly roofs, under hot daytime summer conditions has been growing in recent years using models and simulations[24–26,38–40], microclimate observations[27,41,42], and laboratory studies[30]. Further real-world field studies are warranted to understand RP thermal performance, specifically concerning the interaction between different heat exposure metrics, i.e., surface ($T_{sfc}$), air ($T_{air}$), and mean radiant temperature ($T_{mrt}$). $T_{mrt}$ is the weighted sum of short and longwave radiation that a human experiences at a given place and time, and is considered the most significant heat exposure metric in hot and dry spaces[43,44].

As a daytime heat mitigation strategy with potentially lasting effects throughout the night, it is of particular interest to understand the thermal performance of RP in hot and dry areas where solar radiation is abundant. Numerous questions remain surrounding the effect of RP on localized $T_{sfc}$, $T_{air}$, and $T_{mrt}$ in real-world conditions across different times of day. Surface temperature reduction due to RP has been demonstrated successfully through observations and modeling, but the effect on air temperature is still contested due to the scale of interventions[45] and accurate albedo values applied in models. A recent review of modeling studies found a 0.2–0.6 °C $T_{air}$ reduction

per 0.1 increase in albedo, on average, of the entire neighborhood[46]. However, even if higher albedo results in $T_{air}$ reduction, a trade-off may exist with increased $T_{mrt}$ adversely affecting pedestrians in the daytime resulting from the added reflection of solar radiation towards people[25,47]. However, $T_{mrt}$ is rarely used as a heat exposure metric to quantify the human-experienced impacts of heat mitigation technologies. Different heat exposure metrics impacting the overall thermal load experienced by people have not been compared in neighborhoods that received RP. Moreover, there is minimal empirical research on the impacts of RP on nighttime cooling of $T_{air}$.

This study evaluates a large-scale implementation of RP in the City of Phoenix, Arizona, USA between August and October 2020, where the city applied RP to 58 km of residential neighborhood streets and one public parking lot. The applied RP is a water-based asphalt emulsion seal coat designed to achieve lower pavement surface temperatures on streets through its lighter color and higher albedo. This City-University collaborative project—titled the Cool Pavement Pilot Program (CPPP)—systematically evaluates the performance of the RP to understand its localized heat mitigation potential across extreme heat days based on $T_{sfc}$, $T_{air}$, and $T_{mrt}$. The RP is compared to conventional and commonly used, yet aged, AC sealcoats. The performance evaluation of the RP addresses the following research questions:

1. Compared to weathered/aged AC, how does RP alter $T_{sfc}$, $T_{air}$, and $T_{mrt}$ across four times of day in three neighborhoods during late summer in Phoenix, AZ?
2. How does the surface reflectivity of the RP change over time (7 months) compared to AC in Phoenix, AZ?

We provide innovative neighborhood-scale RP evaluation results, potential impacts on nighttime heat mitigation of RP, possible trade-offs between the use of multiple heat exposure metrics, unintended consequences when using a wrong heat metric, and surface reflection changes over time. Critical recommendations are provided to support optimal location selection of RP based on shade presence and urban form (e.g., height-to-width ratio), as well as human exposure based on time of day, to counteract urban overheating in a hot and dry city using abundant car-centric space that cannot be used for alternative heat mitigation strategies.

## Results
### Heat metrics overview
Weather conditions on data collection days (August 18, September 5, and September 20, 2020) were clear, sunny, and hot, with calm-to-light winds. The daily profiles for mesoscale $T_{air}$, relative humidity (RH), and wind speed from the Phoenix airport are provided in Fig. 1. The $T_{air}$ during the individual measurement transects in the neighborhoods (Fig. 2) was consistent during the pre-sunrise and the afternoon transect, increased during the noon transect, and decreased during the post-sunset transect. The maximum (minimum) $T_{air}$ at Phoenix Sky Harbor airport on these days was 46.1 °C (32.2 °C), 45.6 °C (28.9 °C), and 41.1 °C (25.6 °C) on August 18 (Garfield), September 5 (Maryvale), and September 20 (Westcliff), respectively. All transect measurements occurred at RH levels between 9 and 20% apart from pre-sunrise measurements. Wind speeds were consistently between a calm and moderate breeze (Beaufort scale: 0–4). All 3 days had similar synoptically uninterrupted profiles.

Detailed results comparing each heat exposure metric by neighborhood and time window are provided below and in Figs. 3–5. Overall, significant within-neighborhood differences in $T_{sfc}$ between RP and AC were observed across all four periods. $T_{air}$ differences were within the uncertainty of the instruments between RP-treated and AC areas across the full day. At the same time, $T_{mrt}$ was elevated over RP-coated streets during the noon and afternoon hours compared to AC, and slightly lower at sunrise and sunset.

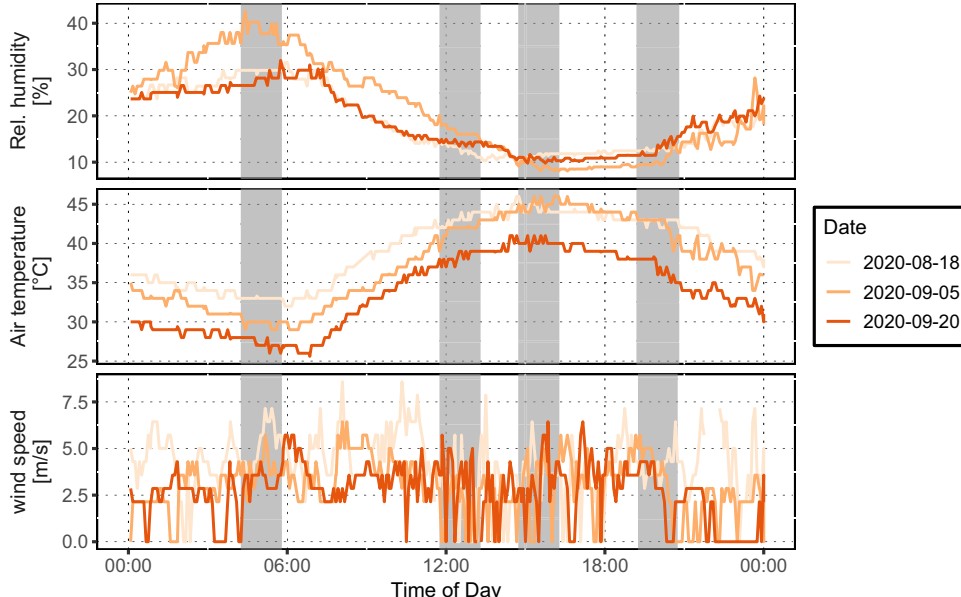

**Fig. 1 | Diurnal meteorological data profile for Phoenix.** Diurnal meteorological data profile from Phoenix Sky Harbor airport for mesoscale meteorological conditions. Relative humidity, air temperature, and wind speed are shown for the 3 days when data were collected (August 18, September 5, and September 20, 2020). Gray highlighted areas denote the time windows during which data collection in the three neighborhoods occurred. Source data are provided as a Source Data file.

## Surface temperature ($T_{sfc}$)

During vehicle traverses across the four in situ measurement periods, the highest mean $T_{sfc}$ of 66.7 °C was found on the AC in the Garfield neighborhood during the afternoon transect (hottest $T_{air}$ time of the day; Fig. 3). At this time and day, the RP reached a mean $T_{sfc}$ of 61.6 °C in Garfield. The minimum $T_{sfc}$ values occurred just before sunrise, with Westcliff—which had shorter days, less intense sunlight, and the lowest average $T_{air}$ (Fig. 1)—showing the lowest average minimum $T_{sfc}$ (28.3 °C for AC, which was 1.3 °C higher than the RP $T_{sfc}$ minimum for Westcliff).

The $T_{sfc}$ values of the RP were, on average, significantly lower than those of AC during all measurements (Fig. 3). The highest $T_{sfc}$ difference between RP and AC (i.e., $T_{sfc,RP} - T_{sfc,AC}$) of −8.4 °C was found in the Westcliff neighborhood during the noon transect for vehicle traverse 1 (and −8.2 °C for vehicle traverse 2), with similarly large differences found during the afternoon transect in the same neighborhood (−7.4 and −7.3 °C for traverses 1 and 2, respectively). The lower $T_{sfc}$ of the RP was evident during the noon and afternoon periods for all neighborhoods, reaching maximum differences of −6.8 °C (noon, traverse 2) and −4.5 °C (afternoon, traverse 2), respectively, in Maryvale and Garfield. $T_{sfc}$ measurements during the noon transect in Garfield indicate a $T_{sfc}$ difference of −5.4 °C. The largest $T_{sfc}$ differences were measured during the noon transect in all neighborhoods. The $T_{sfc}$ differences were lowest, yet significant, before sunrise, with differences ranging from 0.9 to 1.6 °C cooler on the RP across all neighborhoods.

## Air temperature ($T_{air}$)

$T_{air}$ data was collected using 1–3 T-type thermocouples at 2 m height on a vehicle performing two traverses four times a day within the three neighborhoods. The highest mean $T_{air}$ in each neighborhood was found in the afternoon, with 45.5 °C in Garfield over AC, 44.1 °C in Maryvale over RP, and 39.2 °C in Westcliff over AC (Fig. 4). Minimum $T_{air}$ for all neighborhoods occurred before sunrise, with small variations in $T_{air}$ across neighborhoods.

The 2 m $T_{air}$ difference between RP and AC locations (i.e., $T_{air,RP} - T_{air,AC}$) was highest on average, just after sunset, averaging −0.3 °C across neighborhoods (ranging from −0.6 °C to +0.1 °C). Across all neighborhoods and traverses, the $T_{air}$ cooling effect of RP only reached a significant difference of −0.7 °C in the afternoon in Maryvale (traverse 1). Daytime differences averaged −0.2 and −0.1 °C above the RP during noon and afternoon, respectively. Significant warming was found before sunrise during traverse 1 in D5, Maryvale (+0.2 °C).

In summary, the 2 m $T_{air}$ over RP was cooler or equivalent to that over AC after sunset in all neighborhoods. Excluding the pre-sunrise measurements, an insignificantly lower, yet varied 2 m $T_{air}$ (−0.19 °C ± 0.05 °C) was predominantly found over RP compared to AC in all neighborhoods.

## Mean radiant temperature ($T_{mrt}$)

$T_{mrt}$ data were collected with a six-directional net radiometer setup on the mobile MaRTy cart[48] at the pre-defined locations and time windows in all neighborhoods. On average, $T_{mrt}$ was elevated over RP compared to AC during noon and afternoon hours (Fig. 5). The largest $T_{mrt}$ difference between RP and AC ($T_{mrt,RP} - T_{mrt,AC}$) of 5.1 °C was found in the Westcliff neighborhood at noon, showing the elevated $T_{mrt}$ above the RP. $T_{mrt}$ differences were minor before sunrise and after sunset, where AC and RP performed nearly equal.

The highest average $T_{mrt}$ levels were found standing on the RP-coated road in the Garfield neighborhood during the afternoon (74.6 °C), which was the hottest time of the day (Fig. 5). At this time, $T_{mrt}$ was 2.3 °C lower (72.3 °C) on the AC. During the afternoon in Maryvale, $T_{mrt}$ was 1.4 °C higher over RP compared to AC, but $T_{mrt}$ was equally high over the adjacent sidewalk next to RP or AC or when standing on RP. Similar to $T_{sfc}$, minimum $T_{mrt}$ values occurred just before sunrise, with Westcliff showing the lowest average $T_{mrt}$ (−19.0–20.0 °C), approaching $T_{air}$ due to the absence of direct solar radiation. After sunset, $T_{mrt}$ was 0.5 to 1.3 °C cooler over RP due to reduced upwelling longwave radiation compared to AC.

## Surface reflectivity in the solar spectrum

Monthly solar reflectivity measurements with a spectroradiometer were performed between November 2020 and May 2021 at fixed RP locations in all eight neighborhoods (D1-D8) plus one AC control (Asphalt X) location. Figure 6 shows reflectivity results over 7 months (November 2020 to May 2021) across the eight Phoenix Council Districts (each with one neighborhood receiving the RP). On average, the

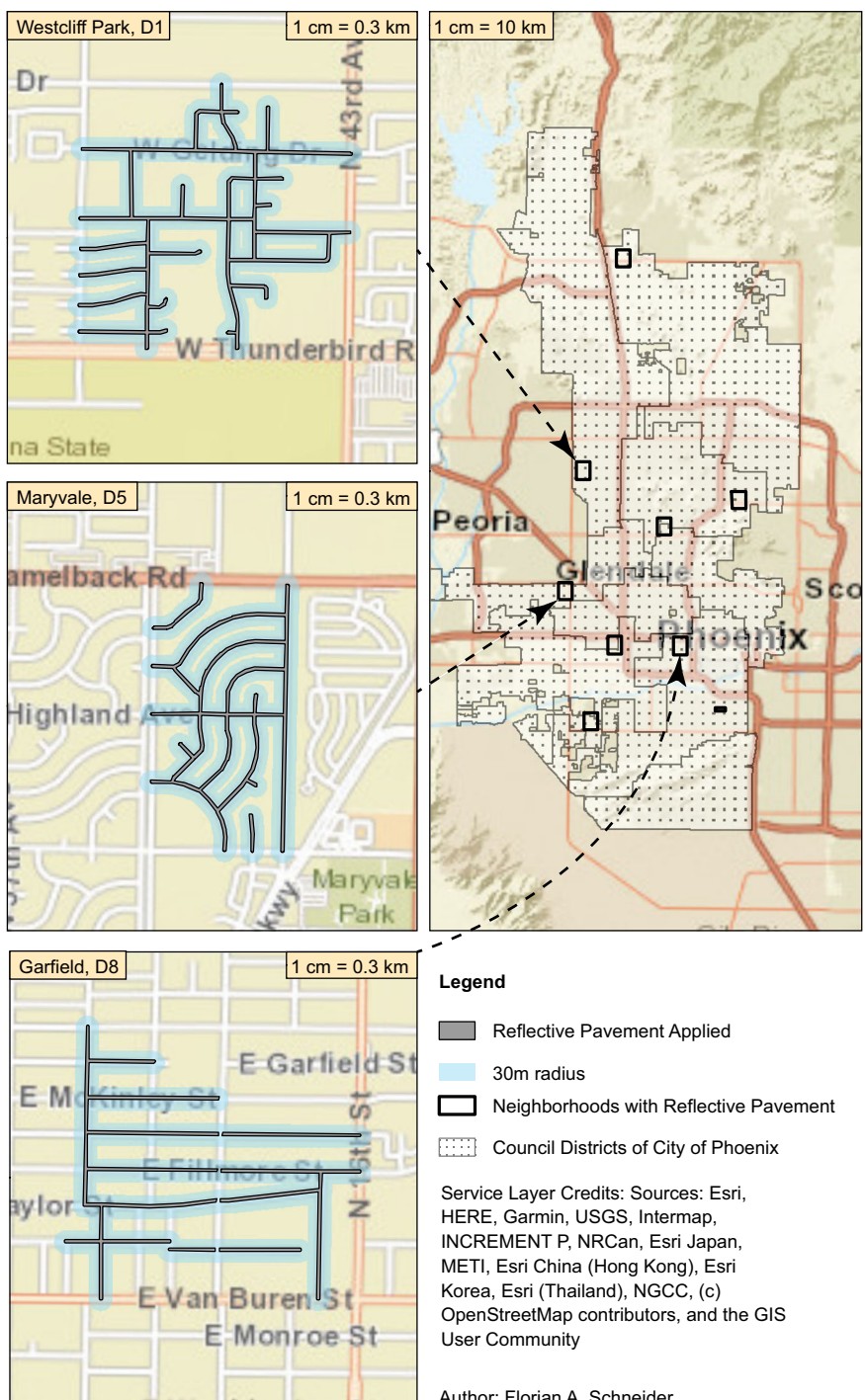

**Fig. 2 | Areas of application of reflective pavement in Phoenix.** Map of the Council Districts (D1–D8) within the City of Phoenix (right), the neighborhoods where CoolSeal was applied (black rectangles), and a close-up for those neighborhoods where in-situ measurements were performed (left). Blue highlighted measured treated roads received the reflective seal treatment. The council district boundaries were downloaded from the open access Phoenix Open Data platform via https://mapping-phoenix.opendata.arcgis.com.

measured treated roads in Council District 3(D3), District 2 (D2), and District 1 southside (D1S) were the most reflective (average reflectivity of 34%, 33%, and 31% of the incident shortwave radiation, respectively). At the same time, RP in District 8 (D8), District 1 north-side (D1N), and District 4 (D4) had the lowest reflectivity (average reflectivity of 24%, 25%, and 28% of the incident shortwave radiation, respectively). These reflectivity values are higher than the average AC reflectivity of ~12–13% in the control segment (Asphalt X, Fig. 6). Throughout the 7 months, all Districts saw decreases in reflectivity (Fig. 6a), with an all-District

average change from 34% to 25% for near-infrared (NIR; 700–2500 nm) and 26% to 18% for visible (VIS; 400–700 nm). These decreases varied by District, where the reflectivity of the surface in D1S, D2, D5, and D6 showed an absolute reduction of 10–12% in 7 months, yet D4, D7, and D8 had an absolute reflectivity decrease of 5–6% across the measured spectrum (350–2500 nm).

Rainfall and street sweeping from December 20–25, 2020 increased the reflectivity in three Districts temporarily (D2, D3, D7), supporting the increase in overall reflectivity in Fig. 6b, yet the

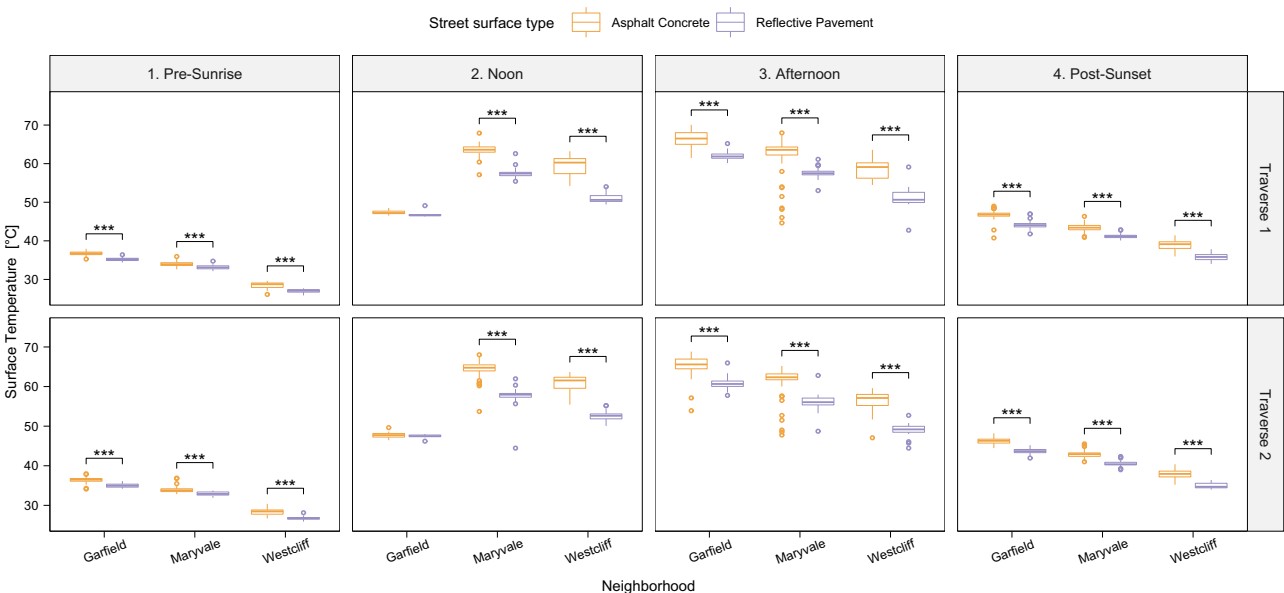

**Fig. 3 | Overview of surface temperature measurements for all areas and times.** Car transect-derived surface temperature panel for all neighborhoods, transects, and traverses in box-whisker plots. Center line = median; box limits = upper and lower quartiles; whiskers = 1.5x interquartile range; points = outliers. Asphalt concrete, AC (orange), and reflective pavement, RP (purple), surface temperature distribution and the statistical significance of the difference between those surface types is shown: \*\*\*$p < 0.001$, \*\*$p < 0.01$, \*$p < 0.05$. Note: The infrared temperature monitor failed during the noon measurements in Garfield, however, data from the MaRTy cart downward facing infrared sensor were used for surface temperature comparisons. Source data are provided as a Source Data file.

remaining Districts were unaffected. Rainfall on March 25, 2021 resulted in increased reflectivity in D4.

## Discussion

### Heat metrics matter

To better understand how RP affects different heat exposure metrics experienced within a residential neighborhood, this study investigated the differences in $T_{sfc}$, $T_{air}$, and $T_{mrt}$ between RP and AC. In-situ comparisons with two mobile measurement platforms were performed four times per day in three residential neighborhoods. $T_{sfc}$ was significantly reduced at all observed times on RP-treated roads compared to AC, while $T_{mrt}$ was reduced after sunset and before sunrise, yet, higher during the noon and afternoon observations.

The significantly lower $T_{sfc}$ indicates that the RP-treated roads absorb less heat than AC roads, which helps to reduce overall urban heat levels and thus should lead to reduced $T_{air}$[49] and $T_{mrt}$ at nighttime due to the lower upwelling infrared radiation from the surfaces. The overall magnitude of the RP $T_{sfc}$ reduction agrees with observations in Los Angeles that showed a moderate $T_{sfc}$ cooling of 4–6 °C for a similar change in albedo[27,42].

While $T_{sfc}$ is reduced significantly by increasing surface solar reflectivity, it comes with a trade-off wherein the radiant heat load is increased due to more solar radiation exposure at the pedestrian level[25,27,47]. The higher $T_{mrt}$ above the RP (middle of road) at noon and afternoon hours is in line with prior modeling[23,28,50] and fieldwork studies[27]. Our results also show that $T_{mrt}$ values on the sidewalk do not differ significantly between RP and AC, but differ significantly when on the road, which is thus a concern for those walking, living, and/or working on the road. The overall solar load (incoming and outgoing) on the human body is the primary contributor to heat stress and thermal discomfort in hot, dry urban microclimates[48]. Given that RP functions best where solar radiation is abundant, this enhanced $T_{mrt}$ will be an ongoing consideration in determining the optimal placement of RP.

Significantly lower radiant heat exposure was found before sunrise, indicating that the lower $T_{sfc}$ leads to a continuously lower $T_{mrt}$ throughout the night in the investigated neighborhoods, though the magnitude of the cooling effect fades over time. Thus, it is essential to

consider the time frame when $T_{mrt}$ could be elevated in already hot environments. One should assess the $T_{sfc}$ and $T_{mrt}$ trade-off based on time of day, pedestrians' behavioral patterns, presence of sidewalks and additional mitigation strategies such as shade[51], and vulnerable populations such as unhoused individuals and outdoor workers. Additionally, it remains unclear how much the additional reflected radiation affects vertical structures, such as buildings next to the street, and whether it could create a higher energy need for indoor cooling[25,52].

We did not observe significant differences in 2 m $T_{air}$ at any time of day averaged over all neighborhoods, indicating no significant neighborhood-scale $T_{air}$ cooling by the RP. Overall, results show minor decreases in $T_{air}$, agreeing with prior modeling studies that found a cooling of 0.2–0.6 °C[46]. Biophysically, a lower $T_{sfc}$, and thus less sensible heat flux to the air volume above the surface, supports this decrease. Reasons for minor $T_{air}$ differences between the RP and AC include a potential displaced cooling effect due to the combination of diffusion and advection, leading to a downwind cooling effect (where no sensors were placed). Additionally, the scale of the intervention (RP treatment) may not be large enough to significantly cool the well-mixed air volume moving over the treated surfaces[45]. Other reasons for small decreases in $T_{air}$ include lack of control for various types of land use throughout the two areas (e.g., grass versus xeriscape), urban design, shading, and irrigation variability across the neighborhood, and more mixing by the time the air reaches 2 m height versus sensing closer to the ground. Furthermore, the sensor uncertainty is large (+/− 1.0 C), potentially obscuring the cooling effect.

Our observational study agrees with prior and current urban climate modeling[46] and the Los Angeles observational studies on single[27] and multiple RP types[42] concerning the overall heat impact of RP use in a neighborhood. This agreement is promising when considering the challenges of accurately measuring or modeling large-scale urban effects across entire neighborhoods.

### Degrading solar reflectivity

Over the 7-month period, monthly measurements indicate that the solar surface reflectivity of the RP reduces from 33–38% to 19–30% across the

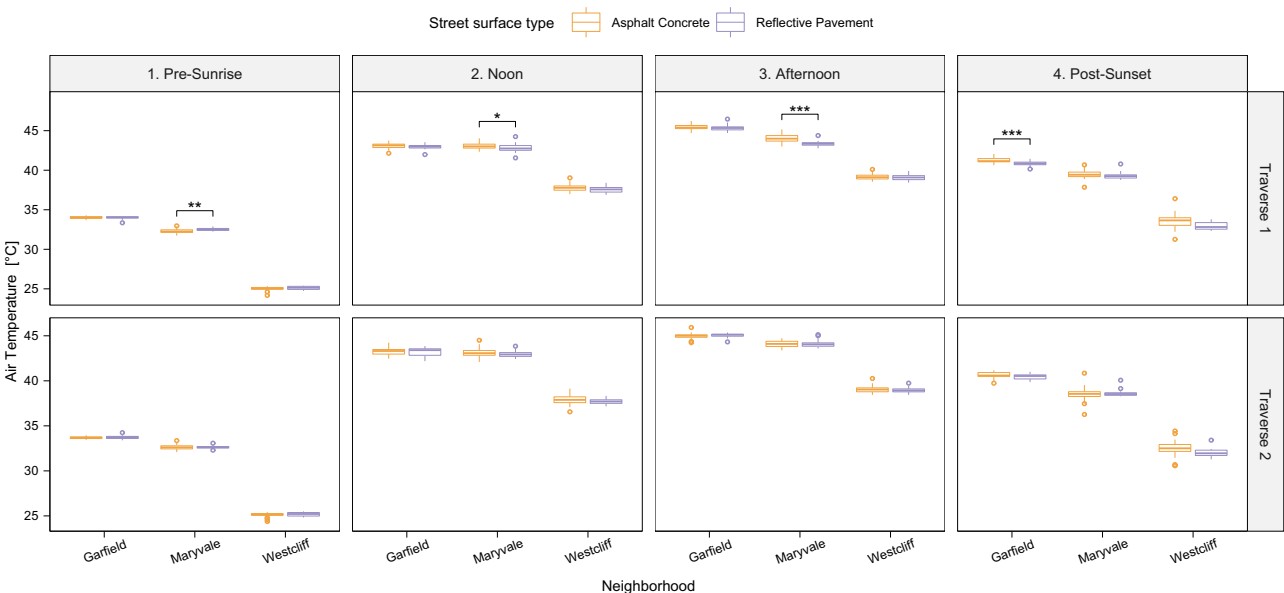

**Fig. 4 | Overview of air temperature measurements for all areas and times.** Car transect-derived air temperature panel for all neighborhoods, transects, and traverses in box-whisker plots. Center line = median; box limits = upper and lower quartiles; whiskers = 1.5x interquartile range; points = outliers. Asphalt concrete, AC (orange), and reflective pavement, RP (purple) surface temperature distribution and the statistical significance of the difference between those surface types is shown: ***$p < 0.001$, **$p < 0.01$, *$p < 0.05$. Source data are provided as a Source Data file.

eight neighborhoods. Similar reflectivity degradation was found by Ko et al. in Los Angeles[42]. These reflectivity reductions may result in a lower cooling effect for $T_{sfc}$ and thus $T_{air}$, yet an improvement for $T_{mrt}$. For comparison, an untreated AC surface had a consistent reflectivity of 12%, reflecting half as much solar radiation as the aged RP surface, and a third as much as the new RP surface. Reduction in reflectivity for RP is likely due to wear, dust accumulation, rubber residual, and other materials[53]. Dust may be a frequent issue for the region where measurements were performed due to regular monsoonal dust storms (haboobs) in Arizona, USA, with a frequency of 9.6 storms per year[54]. After rain and street sweeping events, higher reflectivity values were present, suggesting that cities can use street sweeping to help maintain RP; this benefit of surface cleaning was also observed in a laboratory study[30]. The largest decrease in average reflectivity occurred in neighborhoods with either high traffic volume and/or generally more dust/dirt accumulation. Notably, the reflectivity is highly variable by neighborhood, road, or nearby. For example, the surface reflectivity of locations D1N and D1S differ by -6%, yet are merely 5 m apart. This result is likely due to street design, where the northern side (D1N) is trafficked much more than the street's southern side (D1S), which leads to more rubber residue and wear on the surface.

Fading reflectivity via wear and tear is an important consideration given the demand for more cleaning and maintenance, i.e., to restore higher solar reflectivity[30,42], which helps maintain lower $T_{sfc}$ in the residential neighborhood (this demand may be less in areas that receive more rain and less dust). Notably, the reflectivity results should be interpreted with caution since they only refer to a portion of the roadway (northern side, except location D1S) near the curb that experiences less traffic. Other areas of the road with more traffic may exhibit lower reflectivity. Future work that leverages airborne and spaceborne/satellite imaging spectroscopy can overcome these limitations/drawbacks and provide more spatially-resolved (VSWIR) reflectance data on the variability of the RP reflectance across all neighborhood roads[55,56].

Additionally, the first reflectivity measurements were performed in November, while the RP-treatment was performed between August and October, which could explain aging differences among the different surfaces measured in all neighborhoods. Even so, a statistical

relationship between the age of the surface and its reflectivity could not be found.

## Limitations

In-situ measurements for the three neighborhoods were performed on different dates with slightly different atmospheric conditions (Fig. 1), different RP age, and different reference AC properties. To account for these differences, we compared intra-neighborhood heat exposure metrics for the RP-treated and non-treated sections rather than assessing absolute values. Neighborhood differences were particularly prominent in $T_{sfc}$ results. Although we chose similar micro-environments concerning tree canopy cover and urban morphology in the AC and RP-treated neighborhoods, urban form and vegetation differences still influenced the measurements. Such factors may explain the different magnitudes for the $T_{mrt}$ between RP-treated and non-treated residential neighborhoods.

$T_{air}$ differences between AC and RP-treated neighborhoods were within the sensor uncertainty (+/−1.0 C), which obscures potential cooling effects. The strongest $T_{air}$ cooling is expected close to the RP surface, diminishing as one moves away from the surface, specifically considering turbulence in the urban canopy layer. The turbulent air volume of the urban canopy layer may have restricted us from measuring a significant cooling effect at our measurement height of 2 m, yet the cooling effect may be measurable at lower heights.

## Reflective pavement implementation guidance

Prior to implementing RP, numerous potential co-benefits or trade-offs (e.g., nighttime visibility improvement or morning glare, respectively[24,27]) must be considered based on climate type, land use, urban form and design, road types and speeds, and pedestrian time-use patterns of a specific location. The following recommendations are presented based on current study findings synthesized with available peer-reviewed literature for practical and effective application of RP for heat mitigation:

- RP is most effective in hot mid/low latitude cities with low annual cloud coverage and a large surface area of roads and parking lots, i.e., in low traffic areas, which keep the reflectivity from degrading.

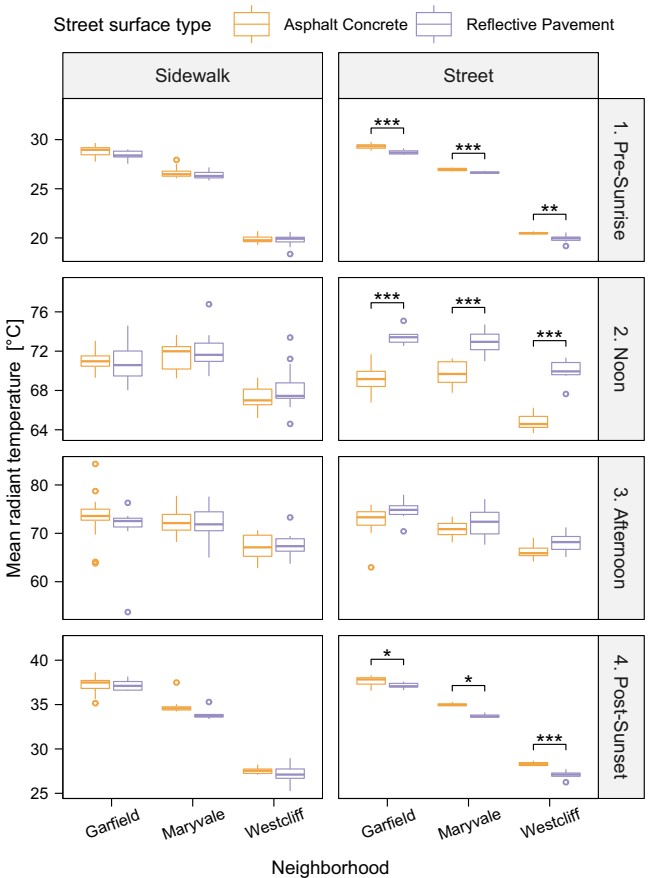

**Fig. 5 | Overview of mean radiant temperature measurements for all areas and times.** Average MaRTy-derived mean radiant temperature (Tmrt) across all neighborhoods and transects, delineated by positions (on adjacent sidewalk (left) versus on road (right)) for asphalt concrete, AC (orange), and reflective pavement, RP (purple) in box-whisker plots. Center line = median; box limits = upper and lower quartiles; whiskers = 1.5x interquartile range; points = outliers. Significance tests denote the difference between the RP and AC surface (\*\*\*$p < 0.001$, \*\*$p < 0.01$, \*$p < 0.05$). Source data are provided as a Source Data file.

- RP is effective in residential neighborhoods with low- to mid-rise buildings and open streets where shade structures, buildings, and trees do not shade the streets; thus, solar radiation can escape the boundary layer when reflected. Trees, overhangs, and canopies may (and should) still be used to provide shading for sidewalks for pedestrian heat stress mitigation. The use of RP cannot replace the benefits of shade trees for pedestrian cooling. RP in urban canyons (high-rise downtown areas) is ineffective for heat mitigation due to a lack of direct incoming solar radiation.
- RP is a convenient and affordable alternative to conventional AC and may extend the life of the pavement (long-term research studies are needed).
- RP should not be used on surfaces with high daytime pedestrian use as it will increase heat load on the body. Those spaces include playgrounds, recreational areas (e.g., basketball courts), courtyards, and plazas. Heat exposure mitigation should focus on shading, such as trees and engineered shade, in these areas.

Optimizing the placement of heat mitigation strategies should consider for whom, what, when, where, and why those strategies will be implemented at a specific location and time[57]. In addition to weighting co-benefits and tradeoffs, community perceptions of a given heat mitigation strategy should also be considered. Determining what the final users—the community experiencing the resulting effects—

think about a given heat mitigation strategy and potential unseen considerations is valuable to city decision-makers. As heat vulnerability is higher for people of color and people below the poverty line[58], who may have faced historical environmental racism, efforts to understand community perception are critical to address heat equity[7], as well as enhance cost-benefit or tradeoff assessments.

This innovative approach of RP addresses areas responsible for added heat in the urban environment and how those can be cooled when urban green spaces are not a viable solution. RP, as a heat mitigation strategy on public property—residential streets—may be an opportunity to be provided equally in all residential neighborhoods, independent of socioeconomic background, because residential streets are prevalent in all neighborhoods of Phoenix, AZ, and implementation does not rely on the wealth of individuals.

## Future challenges

Following this study numerous areas of future research arise. These include:

- Investigating the effect of RP on $T_{air}$ at different heights sensors that have an accuracy that is larger than the expected magnitude of the cooling effect (<1.0 °C).
- Assessing the heat mitigation performance of different RP materials for all heat exposure metrics and the trade-offs of increased $T_{mrt}$ above and adjacent to the surface.
- Examining the effect of additional solar radiation exposure on neighboring vegetation and buildings, which has been called for prior[59] (e.g., the buildings' associated indoor conditions).
- Investigating potential downwind effects of the RP on $T_{air}$, which may be displaced due to the small intervention scale[60], and whether the cooling effects can be enhanced when the scale of the intervention is increased.
- Assessing seasonal, background climate, and local climate zone effects to support multi-model studies for more accurate use in urban planning and city decision-making. Such work may also include understanding the interactions between mixed approaches using multiple heat mitigation strategies[60].
- Investigating the in-situ interaction between trees and RP and how effective RP is in the direct influence area of trees. A modeling study suggests that RP cannot contribute significantly to local urban heat mitigation when a tree or building shades the road[61].

## Reflective pavement assessment

The current study comprehensively assessed RP cooling potential in the hot, dry climate of Phoenix, AZ, considering three heat exposure metrics: $T_{sfc}$, $T_{air}$, and $T_{mrt}$. Based on our findings, we provided context-based suggestions to inform the use of RP to counteract urban over-heating while minimizing unintended consequences.

RP is designed to minimize heat gain in an urban environment in locations where alternative cooling strategies, such as urban green spaces and water features, cannot be placed. Such heat reduction is particularly important for car-centric cities with wide streets and large parking lots. Results comparing untreated to RP-treated neighborhoods show a significant reduction in $T_{sfc}$, no significant impact of RP on $T_{air}$, and mixed results based on time of day concerning $T_{mrt}$ (i.e., significantly elevated around noon, slightly lower at night). These differences may also affect energy use, health, and water usage, yet future work is needed.

The large reduction in $T_{sfc}$ may support a general cooling of the air (shown minimally in the current study), and could thus be used to help counter urban overheating in places where solar radiation is abundant; however, seasonal, background, and local climate effects need to be investigated further to determine the effect and magnitude of RP on the heat exposure metrics when used in other cities and circumstances. The time-of-day $T_{mrt}$ impacts provide important contextual

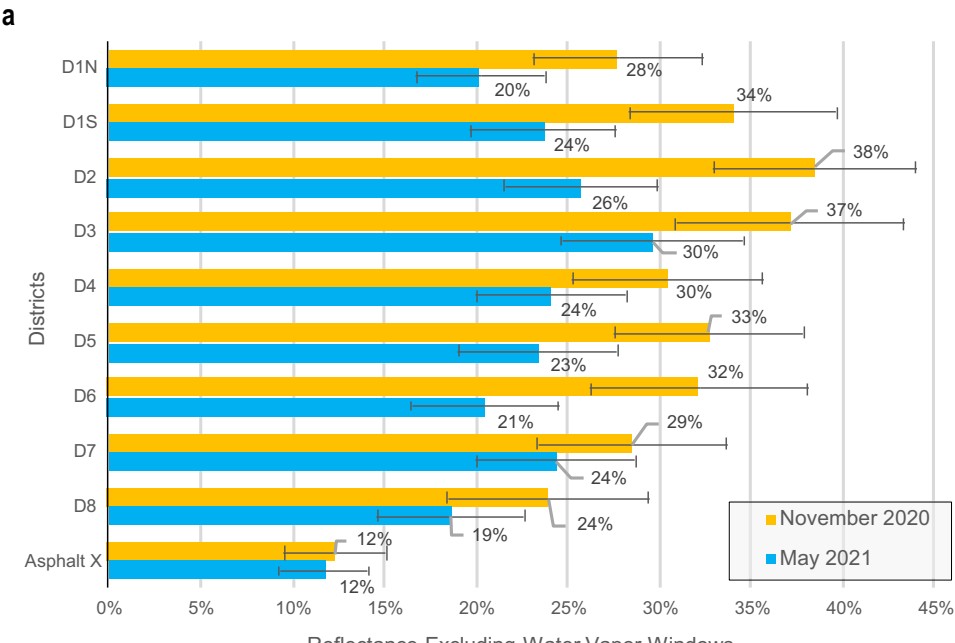

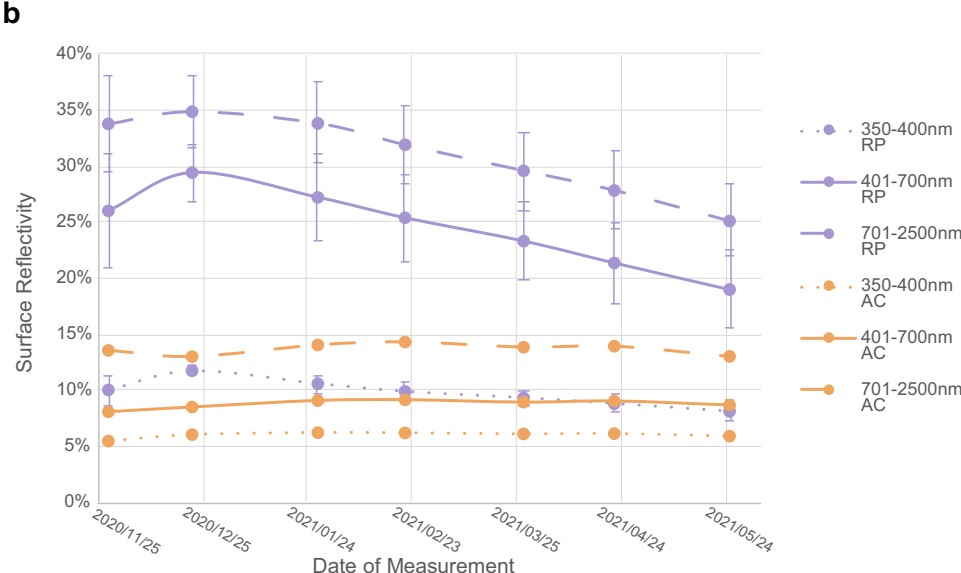

**Fig. 6 | Surface reflectivity measurements in all areas (a) and over time (b).** **a** Reflectivity in November 2020 compared to May 2021 across all wavelengths, excluding strong water vapor and carbon dioxide absorption windows (1350–1450 nm; 1800–1950nm; 2300–2500 nm). **b** Solar reflectivity over time for reflective pavement (RP) and asphalt concrete (AC) across three wavelength ranges: dotted line−ultraviolet A (UVA; 350–400 nm), solid line−visible (VIS; 400–700 nm), and dashed line−near infrared (NIR; 700–2500 nm). RP data represents averages for all 8 Districts, and error bars represent the standard deviation. Source data are provided as a Source Data file. **a** and **b** are modified for colors and readability, with permission, from Figures 11b and 12 of the Cool Pavement Pilot Program report by Middel et al. (2021) published under a CC BY-NC-ND 4.0 license and available at https://hdl.handle.net/2286/R.2.N.160731.

guidance for locational use of RP based on knowledge of the time use of a specific urban space, wherein spaces with high foot traffic midday (parks, plazas, playgrounds) should avoid RP coatings due to heightened $T_{mrt}$ (and thus heat stress) while locations with low foot traffic (roads, parking lots) can provide cooling benefit overnight with RP use. Despite the measurements taken within one local climate zone (open low-rise), similar results are expected in areas with little shade and low cloud coverage−in such locations, the additional reflectivity of RP would result in the greatest changes to the heat exposure metrics tested here, including lasting effects into the night.

Overall, each city must evaluate the potential impact and sustainability of RP to mitigate heat compared to current conditions alongside other strategies. Our synthesis of results and literature provide three important takeaways for cities to consider when assessing the cooling performance of any type of reflective seal. First, the heat metric matters−that is, cooling of the surface occurs throughout most times of the day, yet $T_{mrt}$ is elevated during the day, while $T_{air}$ may see no changes. Second, the location in which RP is applied must ensure that reflected radiation can escape the urban area, and some locations will see greater deterioration (and thus performance decrements) than others. Third, background climate matters, wherein hot, dry, and clear climates will see a greater difference between RP versus conventional seal coats versus warm, humid, more cloudy climates. Finally, RP may not be suitable for

certain locations with high pedestrian foot traffic midday given elevated $T_{mrt}$.

## Methods

### Study area

The City of Phoenix, AZ (33°27'N, 112°04'W) is the capital of and most populous city in Arizona and the heart of the Phoenix metropolitan area (population 4.95 million) in the Southwestern U.S.A. The city has eight Council Districts (Fig. 2), covers an area of 1344.50 km², and experiences a hot desert climate (Köppen Climate Classification subtype Bwh) with, on average, 299 sunny days per year. Summers are hot and dry for 3 months (June, July, and August), with a climatological average (1991–2020) maximum daily $T_{air}$ above 40 °C (104 °F). The area experiences higher humidity during the monsoon season from June 15 to September 30. The average minimum $T_{air}$ from May through September remains above 20 °C (68 °F).

### Study setup

Data were collected to assess (1) the impact of RP on heat metrics across three neighborhoods and (2) long-term solar reflectivity in eight neighborhoods (Fig. 2). All neighborhoods are classified according to Stewart & Oke (2012)[62] as open low-rise local climate zones[63]. In-situ heat measurements were performed in the neighborhoods of Garfield (District 8−D8; 5.6 km RP; 6.5 km AC; 0.8 km Concrete), Maryvale (District 5−D5; 4.5 km RP; 8.0 km AC), and Westcliff Park (District 1−D1; 5.5 km RP; 12.2 km AC).

### In-situ data collection

Three residential in-situ field campaigns collecting heat metric data were conducted on clear-sky, hot days in August and September of 2020 in the Garfield (August 18, 2020), Maryvale (September 5, 2020), and Westcliff (September 20, 2020) neighborhoods (see Fig. 1). Those neighborhoods are in Phoenix Council Districts 8 (D8), 5 (D5), and 1 (D1), respectively. To assess the impact of RP on the urban micro-environment compared to traditional AC in each neighborhood, high-resolution $T_{sfc}$, 2-m $T_{air}$, and $T_{mrt}$ were acquired using two mobile platforms: vehicle ($T_{air}$ and $T_{sfc}$) and human-biometeorological carts ($T_{mrt}$) (see Table 1 for instrument and evaluation details). Data were collected across four 1-h time windows: pre-sunrise (transect to be finished 30 mins before sunrise; ~4:30–5:30), high sun (~12:00–13:00), high air temperature (~15:00–16:00), and directly post-sunset (transect to be started 30 mins after sunset; ~19:30–20:30).

A vehicle was equipped with an Apogee SI-111 Infrared Radiometer positioned on the front of the car perpendicular to the road to monitor $T_{sfc}$ and 1–3 T-type thermocouples to monitor 2-m $T_{air}$; data were recorded at 1 sec intervals on a vehicle moving at ~25 km h⁻¹. This speed allowed the vehicle to complete two traverses of the full neighborhood per hour, stay within speed limit ranges, ensure airflow over the sensors, and to take a representative number of samples in each area. A time-synchronous GPS system was attached to the car and time-matched with each temperature measurements. The movement of the vehicle aspirated the sensors. Traverse measurements while the vehicle was stopped (e.g., at intersections) were excluded from the analysis, as recommended by a similar study[64].

For $T_{sfc}$ measurements, which were taken directly (sensor at 10 cm height) over the AC and RP surfaces, the thermal emissivity of diverse AC surfaces lies between 0.93 and 0.98[65]. Thermal emissivity has a significant effect on $T_{sfc}$, yet a thermal emissivity difference of 0.03 leads to <0.5 °C change in $T_{sfc}$[66]. The infrared radiometer used a default emissivity of 0.95 for all measured surfaces to account for minor thermal emissivity differences. The introduced error is comparable to the error margin of the sensor itself.

A human-biometeorological cart (MaRTy)[48] measured $T_{mrt}$ based on a six-directional net radiometer setup, $T_{air}$, relative humidity, wind velocity, and GPS location at 2-s intervals (Table 1). Within the 1-h traverses, stationary MaRTy measurements occurred at pre-defined locations over and on the sidewalk adjacent to the RP and AC. The cart stopped at each location for 45–60 sec to account for sensor lag[67].

### Long-term reflectivity measurements

Monthly solar reflectivity measurements were performed on clear days between November 2020 and May 2021 at fixed locations in all eight neighborhoods plus one AC control (D3) location. Measurements were taken with an ASD FieldSpec 4 Wide-Res field spectroradiometer and started when RP was 1–3 months old, depending on the neighborhood. This instrument provides solar reflectance data between 350–2500 nm. The monthly dataset provided critical pavement reflectivity performance based on real-world conditions, including seasonal impacts, surface wear, traffic flow and type, and dust and dirt.

Up to ten data points per surface were measured. These measurements were collected on the North side of the road next to the sidewalk to minimize the effect of varying traffic intensities between the neighborhoods, and hence the impact of traffic on road conditions.

### Data analysis/statistics

The mobile temperature observations ($T_{air}$, $T_{sfc}$, and $T_{mrt}$) were time-detrended to account for temporal changes in atmospheric conditions. Time-detrending for MaRTy data ($T_{air}$, $T_{mrt}$) uses a reference location at the start and end of the transect that assumes a linear $T_{air}$ change within the transect hour. Time-detrending of $T_{air}$ and $T_{sfc}$ of the vehicle traverses is based on a microclimate grid-detrending method developed for this study. Microclimate grid-detrending averages $T_{air}$

## Table 1 | List of instruments used to measure each heat metric and the solar reflectivity of the RP

| Instrument | Accuracy | Response Time | Height (m) | Heat metric | Statistical test | Stationary (S) or Mobile (M) |
|---|---|---|---|---|---|---|
| 1–3 T-type Thermocouples (car) | ±1.0 °C | <0.1 s | 2.0 m | $T_{air}$ | Mann-Whitney U test | M |
| Platinum Resistance Thermometer (MaRTy) | ±0.2 °C at (23 °C); ±0.5 °C at (−40/60 °C) | <22 s | 1.7 m | $T_{air}$ | descriptive | S |
| Apogee SI-111 Infrared Radiometer (car) | ±0.2 °C at (−10/65 °C); ±0.5 °C at (−40/70 °C) | <1 s | 0.10 m | $T_{sfc}$ | Mann-Whitney U test | M |
| Downward facing Pyrgeometer – IRO1 (MaRTy) | ±2.4% on $T_{sfc}$ on daily sum | <18 s | 0.99 m | $T_{sfc}$ | descriptive | S |
| 3 NR01 Hukseflux 4-Component Net Radiometers oriented in 6 directions (MaRTy) | ±2.4% on $T_{mrt}$ on daily sum | <18 s | 1.11 m | $T_{mrt}$ | descriptive | S |
| ASD FieldSpec 4 Wide-Res Field Spectroradiometer | 3 nm (VNIR) 30 nm (SWIR) | N/A | 1.0 m | N/A | descriptive | S |

**Table 2 | N-th observation "*n*" and the resulting number of observations "#obs" that could be used in statistical analysis to avoid autocorrelation for each neighborhood, transect hour, and car-derived heat metric**

| Neighborhood | Transect hour | Tair | | | | Tsfc | | | |
|---|---|---|---|---|---|---|---|---|---|
| | | Traverse 1 | | Traverse 2 | | Traverse 1 | | Traverse 2 | |
| | | *n* | #obs | *n* | #obs | *n* | #obs | *n* | #obs |
| Garfield | Pre-sunrise | 44 | 27 | 22 | 58 | 12 | 101 | 11 | 116 |
| | Noon | 35 | 38 | 48 | 24 | 69 | 21 | 36 | 33 |
| | Afternoon | 33 | 35 | 28 | 42 | 19 | 63 | 13 | 90 |
| | Post-sunrise | 25 | 50 | 37 | 35 | 11 | 113 | 16 | 79 |
| Maryvale | Pre-sunrise | 28 | 63 | 25 | 67 | 18 | 98 | 22 | 76 |
| | Noon | 17 | 94 | 22 | 93 | 13 | 122 | 11 | 186 |
| | Afternoon | 19 | 92 | 12 | 99 | 10 | 174 | 9 | 132 |
| | Post-sunrise | 32 | 50 | 25 | 60 | 15 | 106 | 17 | 87 |
| Westcliff | Pre-sunrise | 71 | 28 | 50 | 35 | 33 | 60 | 26 | 67 |
| | Noon | 27 | 67 | 22 | 83 | 30 | 60 | 28 | 65 |
| | Afternoon | 14 | 109 | 13 | 113 | 31 | 50 | 25 | 59 |
| | Post-sunrise | 65 | 25 | 45 | 31 | 29 | 55 | 27 | 51 |

and $T_{sfc}$ data points in an area of 50 m × 50 m during the first and second vehicle traverse (traverse from now on). The difference between the traverse averages of $T_{air}$ or $T_{sfc}$ is used to create a linear temperature difference between each temperature value in the first and second traverse. The heat difference results from additional heating or cooling over time; hence, the results of the two traverses are kept separate. However, the slopes for each $T_{air}$ and $T_{sfc}$ are used to linearly time-detrend the respective temperatures for each grid cell during the first traverse to the average time of the first traverse, and similarly for the second traverse. This processing allows comparing the $T_{air}$ or $T_{sfc}$ across the whole neighborhood incorporating spatial and temporal changes of the heat metrics during measurements. After detrending, the average differences between AC and RP were calculated.

Due to the nature of the continuously collected $T_{air}$ and $T_{sfc}$ data during each traverse and the potential temporal autocorrelation, an autocorrelation test (Durbin-Watson test) for both traverses of each transect hour was applied using microclimate grid-detrended $T_{air}$ (averaged above all 1–3 sensors used) and $T_{sfc}$. We found high temporal autocorrelation, which could artificially inflate statistical power. To minimize the impact, the iterative autocorrelation test method using every n-th observation described in Vanos et al. (2020)[68] was used. After correcting for autocorrelation (*p*-value for both traverses: $p > 0.05$), the n-th observation for both $T_{air}$ and $T_{sfc}$ traverses was identified. The value for n was different for each transect hour, neighborhood, and heat metric. An overview of the autocorrelation parameters is provided in Table 2, which shows the n-th observation used for each transect hour, neighborhood, and heat metric, as well as how many observations for each traverse were available for statistical analysis after adjusting for temporal autocorrelation.

The reduced datasets were tested individually for statistical significance concerning the underlying surface type (AC and RP) using the non-parametric statistical Mann-Whitney U test. All statistical tests and data management were conducted in RStudio version 1.3.1073 (RStudio Team 2020). The DHARMa package was used for the temporal autocorrelation test[69]. The dplyr package was used for the non-parametric statistical Mann–Whitney U test[70]. Statistical significance of differences between surface types is calculated and indicated in Figs. 3–5 with * for statistical significance ($p < 0.05$), ** for good statistical significance ($p < 0.01$), and *** for high statistical significance ($p < 0.001$). Spectroradiometer-derived surface reflectivity was processed by excluding the solar radiation signal for strong water vapor and carbon dioxide absorption windows (1350–1450 nm;

1800–1900nm; and 2300–2500 nm) to prevent very low but strongly varying signals that would introduce strong noise to the data analysis when comparing them to the white surface reference of the spectroradiometer. All data points were then averaged (maximum of 10) for each surface, location, and time measured. In addition to the spectral profiles, reflectance data were grouped into three wavelengths and averaged to show reflectance for particular wavelength spectra: 350–400 nm (UV-A, ultraviolet-A), 400–700 nm (VIS; visible), and 700–2500 nm (NIR; near-infrared). Street sweeping and rainy days were identified in the dataset to determine whether surfaces were cleaned by these processes and thus potentially changed or influenced solar reflection properties. Although no measurements were taken directly after rain or road sweeping, the effects of those events could have a prolonged impact. Due to the low number of data points at each given measurement date, there was low power to perform statistical testing; thus, descriptive statistics are provided between the different RP-treated neighborhoods and AC.

## Data availability

The datasets generated during and/or analyzed during the current study are available in the DesignSafe-CI repository with the identifiers for the heat exposure metric data https://doi.org/10.17603/ds2-71a1-n812[71] and the surface reflectivity data https://doi.org/10.17603/ds2-a1nj-z717[72]. Source data are provided with this paper. Some data in this manuscript have been preliminarily disclosed in a report available at https://hdl.handle.net/2286/R.2.N.160731. Source data are provided with this paper.

## Code availability

Code to reproduce the figures is available upon request.

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

## Acknowledgements

The authors of this paper are grateful for financial support from the City of Phoenix, Project no. ST87400360 (A.M. and J.K.V.) and from the Healthy Urban Environment Initiative (A.M. and J.K.V.), which is grateful for the support that it has received from the Maricopa County Industrial Development Authority (MCIDA) Award #AWD00033817. We would like to acknowledge all the volunteers that were essential in the data collection process during hazardous extreme heat events.

## Author contributions

F.A.S., J.K.V., and A.M conceptualized the study. J.K.V. and A.M. acquired the funding and supervised the project. Data collection fieldwork was led and supervised by F.A.S. Data was collected by F.A.S, J.C.O., J.K.V., D.J.S., and A.M. The analysis was conducted by F.A.S. Figures were produced by F.A.S. with help from J.C.O. Results were critically assessed and interpreted by F.A.S., J.C.O., J.K.V., D.J.S., and A.M. The manuscript draft was written by F.A.S. All authors (F.A.S., J.C.O., J.K.V., D.J.S., and A.M.) edited and approved the manuscript.

## Competing interests

The authors declare no competing interests.
