## [Peer Review File · Nature Communications]

Evidence-based guidance on reflective pavement for urban heat mitigation in ArizonaREVIEWER COMMENTS

Reviewer #1 (Remarks to the Author):

The paper has collected and plotted surface, air, and mean radiant temperature over some high-albedo pavements.

The instruments used for measuring surface and air temperature are non-research grades as the authors have clearly stated themselves in the section titled 'future challenges': "investigating the effect of RP on Tair at different heights with more precise sensors." As such it is not clear if the measurements are accurate and what the 'data' really represents. Measuring through air temperature requires precision instrument in an aspirated system with statistically representative samples.

The same is through with the measurement of surface temperatures with infrared sensors. For such measurements, one has to assume a thermal emissivity for the surfaces.

The globe measurements of the mean radiant indicates radiation incident on the target. Having a light-colored target, for sure, minimizes the absorption of the short-wave radiation. However, when radiation is absorbed by the surface and emitted back as long-wave radiation will be absorbed by the target. There is no discussion of radiation energy balance on the globe meter.

The paper concludes that the MRT on the reflective pavement increases significantly. The air temperature does not change much. So the question is how the surface energy is balanced. The paper does not have any analysis of energy balance on the surface (both instantaneous and diurnal).

Most conclusions are general and are not the results of the research presented in the paper.

Some terminologies need to be scientifically defined, e.g., what is the meaning of 'urban overheating'. How is it measured?

The introduction of the paper has many non relevant materials and can be easily omitted.

So, in summary, this reviewer is left with a paper discussing some perhaps questionable data, no analysis of the data (except plotting them), and some general conclusions that are not based on the research presented in the paper.

Reviewer #2 (Remarks to the Author):

The paper provides extensive experimental results on the contribution of reflective pavements for roads and parking lots to counter the effects of urban heat island. More specifically, three heat exposure metrics are considered, showing advantages and limitations of reflective pavements in the considered urban environment. While not providing outstandingly new results, the paper supports recent findings and gives a clear view on the subject. In this reviewer's opinion, it can be considered for publication in its current form.

Minor suggestions to improve the paper:

In line 48, I'd write heat exposure metric of interest.

In line 52, please develop the contribution of different references cited as (1-5). This could be useful also elsewhere in the followings.

In Fig. 3, all data are marked *** (<0.001), while different values occurs in the following figures. The explanation on the statistical significance of the differences could be moved in the text.

Below Fig. 6, or somewhere else (e.g. where the used instrument is detailed), it should be explained why the water vapor and carbon dioxide absorption windows have been excluded.

In line 415, check Tmrt.

In line 487, Tmrt instead of TMRT.

Reviewer #3 (Remarks to the Author):

Review of Evidence-Based Guidance on Reflective Pavement for Urban Heat Mitigation: A Case Study in Phoenix, Arizona by Schneider et al.

Recommendation: Minor revisions

General comments: The objectives of the study were on evaluating a large-scale implementation of reflective pavement (RP) seal application in the City of Phoenix over several urban land cover types, with the intention of evaluating how RP compares with asphalt concrete (AC) in cooling several meteorological variables over a 7 month period (pg 6, L128-142).

- The study is very clearly written and easily accessible to both an academic and urban policy making audience who would be interested in the efficacy of this heat mitigation approach, especially in rapidly-growing subtropical desert cities in which the study is based in. The results are discussed in the appropriate context, with the trade-off between T_{sfc} and radiant heat between RP and AC being a key result. I have no doubt that this paper will be highly topical to multi-disciplinary researchers in urban climate or urban planning.

-The method is sound and clearly described. In particular, the novel use of micrometeorological carts that directly measure at the appropriate meteorological scale is a robust approach to obtain very relevant data. Data analysis is also robustly done, and the figures are clear for comprehension.

- I see no reason why this paper should not be published in this journal, pending two minor issues worth addressing.

1) L283-322: There appears to be no mention of the potential impacts of regular AZ monsoonal dust storms/haboobs (e.g. Eagar et al. 2017, <https://www.sciencedirect.com/science/article/abs/pii/S1875963716300969>) on degradation of solar reflectivity. Some discussion on the impacts of these acute events to maintenance of RP should be included.

2) Consistency in the use of RP and AC throughout the paper should be checked by the authors. e.g. pg 6, L138-140 - both RP and AC were defined in the preceding pages. The captions of figures -- why is RP/AC used on Fig 6, but completely spelled out in the other figures?

REVIEWER COMMENTS (responses and newly added references in BLUE)

Reviewer #1 (Remarks to the Author):

The paper has collected and plotted surface, air, and mean radiant temperature over some high-albedo pavements. The instruments used for measuring surface and air temperature are non-research grades as the authors have clearly stated themselves in the section titled 'future challenges': "investigating the effect of RP on Tair at different heights with more precise sensors." As such it is not clear if the measurements are accurate and what the 'data' really represents. Measuring through air temperature requires precision instrument in an aspirated system with statistically representative samples.

We think that Reviewer #1 may have misunderstood the methodology. All instruments are research-grade. The instruments to measure air temperature at 2-m (± 1.0 deg C for the car traverses, 0.1 deg C for the mobile cart traverses) and surface temperature (± 0.2 deg C) are research-grade. The full biometeorological cart used professionally calibrated, gold-standard Campbell Scientific instrumentation. The specs and information on all are provided in the methods of the paper (see Table 1). All sensors were NIST calibrated. In addition, observations from multiple instruments were compared to each other and showed statistical effects based on hour and location. Related to the mobile transects, the Apogee surface temperature sensor has a low error margin $\pm 0.2^\circ\text{C}$ at $(-10/65^\circ\text{C})$.

We recognize that our language about "Tair at different heights with more precise sensors" was misleading and incorrect. We thank reviewer #1 for pointing this out. Clarification: The segment on "future challenges" that Reviewer 1 refers to is not methods – it is about future directions for improving the understanding of reflective pavement impacts on air temperature at different heights (we used one height with multiple sensors in the present study). The suggested future work looking at air temperature gradients refers to the vertical position rather than sensor precision. The air temperature sensors used in this study for the car-measurements have an error-margin of 1.0 deg C and similar instruments, though with lower error margin of 0.15 deg C, have been used in prior studies to investigate neighborhood scale effects via vehicle traverses ¹. Hence, we suggest improving the error margin in the future, which is what we meant by writing "Tair at different heights with more precise sensors". We adjusted the text to improve clarity and added a clarification that the system was aspirated.

Text at lines 478–481 reads as follows:

"A time-synchronous GPS system was attached to the car and time-matched with temperature measurements. The movement of the vehicle aspirated the sensors. Traverse measurements while the vehicle was stopped (e.g., at intersections) were excluded from the analysis, as recommended by a similar study ¹."

The results of air and surface temperature from the car traverses agree with the measurements of the research- and gold-standard for mobile measurements with MaRTy ($> \$20,000$ instrument platform) ^{2,3} that were taken at the same time in the same area. The instrument is referred to in our instrument list (Table 1).

The same is through with the measurement of surface temperatures with infrared sensors. For such measurements, one has to assume a thermal emissivity for the surfaces.

Thank you for noting the importance of emissivity. The thermal emissivity of diverse asphalt surfaces lies between 0.93 and 0.98 ⁴. Thermal emissivity has a significant effect on the surface temperature, yet a thermal emissivity difference of 0.03 leads (assuming an emissivity of 0.95 with the sensor) to less than 0.5 deg C change in surface temperature ⁵. The difference of 0.93 to 0.98 for the emissivity is considered a marginal error when considering the significant temperature differences (The T_{sfc} differences were lowest, yet significant, before sunrise, with differences ranging from 0.9 to 1.6°C cooler on the RP across all neighborhoods; any other difference was higher and up to 8.4°C during the noon transect in the Westcliff neighborhood) between the asphalt-concrete and reflective coating surfaces. We have adjusted and added the following content to clarify:

Text at lines 482–487 reads as follows:

“For T_{sfc} measurements, the thermal emissivity of diverse AC surfaces lies between 0.93 and 0.98 ⁴. Thermal emissivity has a significant effect on T_{sfc} , yet a thermal emissivity difference of 0.03 leads to less than 0.5°C change in T_{sfc} ⁵. The infrared radiometer used a default emissivity of 0.95 for all measured surfaces to account for minor thermal emissivity differences. The introduced error is comparable to the error margin of the sensor itself.”

The globe measurements of the mean radiant indicates radiation incident on the target. Having a light-colored target, for sure, minimizes the absorption of the short-wave radiation. However, when radiation is absorbed by the surface and emitted back as long-wave radiation will be absorbed by the target. There is no discussion of radiation energy balance on the globe meter.

We did not perform globe measurements in this study. Our instrument list is provided in the publication, and for your reference below. Mean radiant temperature was measured using a state-of-the-art and sophisticated research-grade 6-directional net radiometer setup, MaRTy ^{2,3}.

Table 1. List of instruments used to measure each heat metric and the solar reflectivity of the RP.

Instrument	Accuracy	Response Time	Height (m)	Heat Metric	Statistical Test	Stationary (S) or Mobile (M)
1–3 T-type Thermocouples (car)	$\pm 1.0^{\circ}\text{C}$	$< 0.1\text{ s}$	2.0 m	T_{air}	Mann-Whitney U test	M
Platinum Resistance Thermometer (MaRTy)	$\pm 0.2^{\circ}\text{C}$ at (23°C); $\pm 0.5^{\circ}\text{C}$ at (-40/60°C)	$< 22\text{ s}$	1.7 m	T_{air}	descriptive	S
Apogee SI-111 Infrared Radiometer (car)	$\pm 0.2^{\circ}\text{C}$ at (-10/65°C); $\pm 0.5^{\circ}\text{C}$ at (-40/70°C)	$< 1\text{ s}$	0.10 m	T_{sfc}	Mann-Whitney U test	M
Downward facing Pyrgeometer – IR01 (MaRTy)	$\pm 2.4\%$ on T_{sfc} on daily sum	$< 18\text{ s}$	0.99 m	T_{sfc}	descriptive	S

3 NRO1 Hukseflux 4-Component Net Radiometers oriented in 6 directions (MaRTy)	$\pm 2.4\%$ on T_{mrt} on daily sum	< 18 s	1.11 m	T_{mrt}	descriptive	S
ASD FieldSpec 4 Wide-Res Field Spectroradiometer	3 nm (VNIR) 30 nm (SWIR)	N/A	1.0 m	N/A	descriptive	S

The paper concludes that the MRT on the reflective pavement increases significantly. The air temperature does not change much. So the question is how the surface energy is balanced. The paper does not have any analysis of energy balance on the surface (both instantaneous and diurnal).

The manuscript focuses on above-ground thermal impacts of reflective pavement that may impact decision-making. Surface energy balance measurements, including sensible and latent heat flux, are not core to this work.

Most conclusions are general and are not the results of the research presented in the paper.

All conclusions are a result of our presented research. Our work connects different heat metrics, their effects on the environment and people, resulting in recommendations where reflective pavement is most effective after deployment. Such conclusions are vital for decision makers. The new findings are based on statistically significant heat metric results and short-term wear effects on the reflective pavement with respect to surface albedo. Thus, the study creates useful interdisciplinary information beyond the scientific literature universe and realistically addresses concerns and benefits of reflective pavement, which we feel is portrayed in the conclusions.

Some terminologies need to be scientifically defined, e.g., what is the meaning of 'urban overheating'. How is it measured?

Thank you for bringing up urban overheating terminology. Urban overheating is a concept recently introduced ⁶. It not a measurable unit, but rather highlights the intra-urban variability of heat and its compounding effects and sources. Specifically, urban overheating represents a multifaceted threat to the well-being, performance, and health of individuals as well as the energy efficiency and economy of cities. It is influenced by complex interactions between building, city, and global scale climates (also referred to in the present manuscript). Intra-urban heat differences (which are measured in the current paper) can be measured by identifying much more localized (neighborhood-scale) heat and cool island as well as including the socio-economic factors of the local population.

Text at line 54 reads as follows:

“Urban overheating, a recently introduced concept...”

The introduction of the paper has many non relevant materials and can be easily omitted.

Seeing the meteorological results as isolated disciplinary outcomes will not assist with the future development of innovative heat mitigation strategies and where they can be placed. Our study is intentionally placed within the interdisciplinary environment. We must consider the holistic context and audience that our study addresses. Our study is not only a report on heat metric measurements. It is a synthesis with knowledge from other interdisciplinary areas with measurements and analysis based on the needs of decision makers within cities and experts in the field. It thus expands from heat and reflectivity metrics to the application and suggestions for decision-makers of urban design. Our outcomes also identify holistic factors that must be considered beyond a pure micro-climatological perspective by synthesizing the results with background climate, population vulnerability, and considering generally aspects of marginalized communities in urban development.

So, in summary, this reviewer is left with a paper discussing some perhaps questionable data, no analysis of the data (except plotting them), and some general conclusions that are not based on the research presented in the paper.

Thank you for your time to review our paper in detail and provide useful feedback to improve our publication's outcomes and readability.

Reviewer #2 (Remarks to the Author):

The paper provides extensive experimental results on the contribution of reflective pavements for roads and parking lots to counter the effects of urban heat island. More specifically, three heat exposure metrics are considered, showing advantages and limitations of reflective pavements in the considered urban environment. While not providing outstandingly new results, the paper supports recent findings and gives a clear view on the subject. In this reviewer's opinion, it can be considered for publication in its current form.

Thank you for the summary and for approving the manuscript in its current form. We present reproducible data that confirms other studies, synthesize recent findings, and provide an interdisciplinary perspective on recommended applications.

Minor suggestions to improve the paper:

In line 48, I'd write heat exposure metric of interest.

That is a great suggestion; changed.

In line 52, please develop the contribution of different references cited as (1-5). This could be useful also elsewhere in the followings.

We have clarified the reference contribution by changing the content:

Text at lines 50–53 reads as follows:

“Cities globally have experienced elevated temperatures due to anthropogenic heat sources and the built environment ⁷, which result in an Urban Heat Island (UHI) by adding or retaining more energy within the urban system ^{8,9}. This urban-induced warming together with climate change result in increasing temperatures in cities ¹⁰.”

In Fig. 3, all data are marked *** (<0.001), while different values occurs in the following figures. The explanation on the statistical significance of the differences could be moved in the text.

Thank you. We have added the statistical significance to the captions to allow them to be standalone. We adjusted and added the explanation on the statistical significance to the text within our Methods/Data Analysis section:

Text at lines 540–543 reads as follows:

“Statistical significance of differences between surface types is calculated and indicated in Figures 3–5 with * for statistical significance ($p < 0.05$), ** for good statistical significance ($p < 0.01$), and *** for high statistical significance ($p < 0.001$).”

Below Fig. 6, or somewhere else (e.g. where the used instrument is detailed), it should be explained why the water vapor and carbon dioxide absorption windows have been excluded.

Thank you for asking for this clarification. It was a hidden assumption. We have adjusted and added the following explanation:

Text at lines 544–549 reads as follows:

“Spectroradiometer-derived surface reflectivity was processed by excluding the solar radiation signal for strong water vapor and carbon dioxide absorption windows (1350–1450nm; 1800–1900nm; and 2300–2500nm) to prevent very low but strongly varying signals that would introduce strong noise to the data analysis when comparing them to the white surface reference of the spectroradiometer. All data points were then averaged (maximum of 10) for each surface, location, and time measured.”

In line 415, check Tmrt.

In line 487, Tmrt instead of TMRT.

Thank you for finding those typos.

Reviewer #3 (Remarks to the Author):

Review of Evidence-Based Guidance on Reflective Pavement for Urban Heat Mitigation: A Case Study in Phoenix, Arizona by Schneider et al.

Recommendation: Minor revisions

General comments: The objectives of the study were on evaluating a large-scale implementation of reflective pavement (RP) seal application in the City of Phoenix over several urban land cover types, with the intention of evaluating how RP compares with asphalt concrete (AC) in cooling several meteorological variables over a 7 month period (pg 6, L128-142).

Thank you for the summary. We only measured the three heat metrics within a 1 month window across three full day field campaigns. Monthly (one time per month) measurements were taken over 7 months. Below are the texts that refer to the measurements. We have slightly adjusted them to ensure clarity for readers.

Text at lines 460–462 reads as follows:

“Three residential in-situ field campaigns collecting heat metric data were conducted on clear-sky, hot days in August and September of 2020 in the Garfield (August 18, 2020), Maryvale (September 5, 2020), and Westcliff (September 20, 2020) neighborhoods (see Figure 1).”

Text at lines 494–495 reads as follows:

“Monthly solar reflectivity measurements were performed on clear days between November 2020 and May 2021 at fixed locations in all eight neighborhoods plus one AC (D3) location.”

- The study is very clearly written and easily accessible to both an academic and urban policy making audience who would be interested in the efficacy of this heat mitigation approach, especially in rapidly-growing subtropical desert cities in which the study is based in. The results are discussed in the appropriate context, with the trade-off between T_{sfc} and radiant heat between RP and AC being a key result. I have no doubt that this paper will be highly topical to multi-disciplinary researchers in urban climate or urban planning.

Thank you for commending us on the readability for multiple audiences.

-The method is sound and clearly described. In particular, the novel use of micrometeorological carts that directly measure at the appropriate meteorological scale is a robust approach to obtain very relevant data. Data analysis is also robustly done, and the figures are clear for comprehension.

Thank you.

- I see no reason why this paper should not be published in this journal, pending two minor issues worth addressing.

Thank you.

1) L283-322: There appears to be no mention of the potential impacts of regular AZ monsoonal dust storms/haboobs (e.g. Eagar et al. 2017, <https://www.sciencedirect.com/science/article/abs/pii/S1875963716300969>) on degradation of solar

reflectivity. Some discussion on the impacts of these acute events to maintenance of RP should be included.

Thank you for this suggestion. We have briefly mentioned wear and tear of RP material and how dust, tire tracks, and dirt are potentially responsible for a reduced albedo over time but we agree that a clearer statement with respect to Haboobs is necessary. We added the following discussion content:

Text at lines 305–306 reads as follows:

Dust may be a frequent issue for the region where measurements were performed due to regular monsoonal dust storms (haboobs) in Arizona, USA with a frequency of 9.6 storms per year¹¹.

2) Consistency in the use of RP and AC throughout the paper should be checked by the authors. e.g. pg 6, L138-140 - both RP and AC were defined in the preceding pages. The captions of figures -- why is RP/AC used on Fig 6, but completely spelled out in the other figures?

Thank you for catching that. We fixed the acronym use in the paper.

References

1. Hart, M. A. & Sailor, D. J. Quantifying the influence of land-use and surface characteristics on spatial variability in the urban heat island. *Theor. Appl. Climatol.* **95**, 397–406 (2009).
2. Middel, A. & Krayenhoff, E. S. Micrometeorological determinants of pedestrian thermal exposure during record-breaking heat in Tempe, Arizona: Introducing the MaRTy observational platform. *Sci. Total Environ.* **687**, 137–151 (2019).
3. Middel, A., AlKhaled, S., Schneider, F. A., Hagen, B. & Coseo, P. 50 Grades of Shade. *Bull. Am. Meteorol. Soc.* 1–35 (2021) doi:10.1175/bams-d-20-0193.1.
4. Marchetti, M. *et al.* Emissivity Measurements of Road Materials. *QIRT J.* **1**, (2004).
5. Gui, J., Phelan, P. E., Kaloush, K. E. & Golden, J. S. Impact of pavement thermophysical properties on surface temperatures. *J. Mater. Civ. Eng.* **19**, 683–690 (2007).
6. Nazarian, N. *et al.* Integrated Assessment of Urban Overheating Impacts on Human Life. *Earth's Future* vol. 10 (2022).
7. Taha, H. Urban climates and heat islands: Albedo, evapotranspiration, and anthropogenic heat. *Energy Build.* **25**, 99–103 (1997).
8. Oke, T. R. The energetic basis of the urban heat island. *Q. J. R. Meteorol. Soc.* **108**, 1–24 (1982).
9. Oke, T. R., Johnson, G. T., Steyn, D. G. & Watson, I. D. Simulation of surface urban heat islands under 'ideal' conditions at night part 2: Diagnosis of causation. *Boundary-Layer Meteorol.* **56**, 339–358 (1991).
10. Georgescu, M., Morefield, P. E., Bierwagen, B. G. & Weaver, C. P. Urban adaptation can roll back warming of emerging megapolitan regions. *Proc. Natl. Acad. Sci. U. S. A.* **111**, 2909–2914 (2014).
11. Eagar, J. D., Herckes, P. & Hartnett, H. E. The characterization of haboobs and the deposition of dust in Tempe, AZ from 2005 to 2014. *Aeolian Res.* **24**, 81–91 (2017).

REVIEWERS' COMMENTS

Reviewer #1 (Remarks to the Author):

Thanks for your responses to my comments.

Please note the following:

1. Measurements of air temperature with a non-aspirated instrument can be highly affected by the radiation environment (even in a radiation shielded casings) and can have an error exceeding 5K.
2. Measurement of surface temperature needs an accurate estimate of surface thermal emittance. Most non-metallic urban surfaces have a thermal emittance of 0.80 to 0.95. That variation in the thermal emittance can introduce a error of about 13K for a surface temperature of 300K.
3. Practically all the conclusions of this paper is independent of the study carried out. The novelty of the paper is nil.

Reviewer #2 (Remarks to the Author):

The authors seem to have replied satisfactorily to the reviewvers' remarks. Therefore, the paper can be published in its current form.

Reviewer #3 (Remarks to the Author):

I am satisfied with the changes made in the revision, which has strengthened the paper. I opine it is ready for publication in the journal.

REVIEWER COMMENTS (responses and newly added references in BLUE)

Reviewer #1 (Remarks to the Author):

Measurements of air temperature with a non-aspirated instrument can be highly affected by the radiation environment (even in a radiation shielded casings) and can have an error exceeding 5K.

We thank reviewer #1 for their input. We only used air temperature measurements when the car was moving, so there was natural aspiration for the T-type thermocouples. This is addressed in lines 480 to 488 of our manuscript.

Measurement of surface temperature needs an accurate estimate of surface thermal emittance. Most non-metallic urban surfaces have a thermal emittance of 0.80 to 0.95. That variation in the thermal emittance can introduce a error of about 13K for a surface temperature of 300K.

We thank reviewer #1 for their input and appreciate the clarification and urgency on the matter of emissivity in the context of surface temperature measurements. We have added a statement that the surface temperature sensor is only reading asphalt concrete and reflective pavement surfaces and no other urban fabrics. As addressed in our paragraph from lines 489 to 494, we acknowledge the error margin based on our emissivity assumption for the emissivity ranges of asphalt concrete (AC) surfaces.

Text at lines 489–494 reads as follows:

“For Tsfc measurements, which were taken directly (sensor at 10 cm height) over the AC and RP surfaces, the thermal emissivity of diverse AC surfaces lies between 0.93 and 0.9865. Thermal emissivity has a significant effect on Tsfc, yet a thermal emissivity difference of 0.03 leads to less than 0.5°C change in Tsfc66. The infrared radiometer used a default emissivity of 0.95 for all measured surfaces to account for minor thermal emissivity differences. The introduced error is comparable to the error margin of the sensor itself.”

Practically all the conclusions of this paper is independent of the study carried out. The novelty of the paper is nil.

Thank you for your time to review our paper in detail and provide useful feedback to improve our publication’s outcomes and readability.

Reviewer #2 (Remarks to the Author):

The authors seem to have replied satisfactorily to the reviewers' remarks. Therefore, the paper can be published in its current form.

Thank you for your time to review our paper in detail and provide useful feedback to improve our publication’s outcomes and readability.

Reviewer #3 (Remarks to the Author):

I am satisfied with the changes made in the revision, which has strengthened the paper. I opine it is ready for publication in the journal.

Thank you for your time to review our paper in detail and provide useful feedback to improve our publication’s outcomes and readability.